

# Chemical composition, antioxidant and anti-inflammatory properties of *Monarda didyma* L. essential oil

Daniele Fraternale[1], Hanh Dufat[2], Maria Cristina Albertini[1], Chouaha Bouzidi[2], Rossella D'Adderio[1], Sofia Coppari[1], Barbara Di Giacomo[1], Davide Melandri[3], Seeram Ramakrishna[4] and Mariastella Colomba[1]

[1] Department of Biomolecular Sciences, University of Urbino, Urbino, PU, Italy
[2] Produits Naturels, Analyse et Synthèse, CITCOM-UMR CNRS 8038—Faculté de Santé, Pharmacie, Université Paris Cité, Université de Paris, Paris, France
[3] U. Burns Center, Dermatology and Emilia Romagna Regional Skin Bank, M. Bufalini Hospital, Cesena, FC, Italy
[4] Center for Nanofibers and Nanotechnology, National University of Singapore, Singapore

## ABSTRACT

In the present study, *Monarda didyma* L. essential oil (isolated from the flowering aerial parts of the plant) was examined to characterize its chemotype and to evaluate, in addition to the quali-quantitative chemical analysis, the associated antioxidant and anti-inflammatory activities. The plants were grown in central Italy, Urbino (PU), Marche region. Different analyses (TLC, GC-FID, GC-MS and $^1$H-NMR) allowed the identification of twenty compounds among which carvacrol, p-cymene and thymol were the most abundant. On this basis, the chemotype examined in the present study was indicated as *Monarda didyma* ct. carvacrol. The antioxidant effect was assessed by DPPH assay. Moreover, this chemotype was investigated for the anti-inflammatory effect in an *in vitro* setting (*i.e.*, LPS-stimulated U937 cells). The decreased expression of pro-inflammatory cytokine IL-6 and the increased expression of miR-146a are suggestive of the involvement of the Toll-like receptor-4 signaling pathway. Although further studies are needed to better investigate the action mechanism/s underlying the results observed in the experimental setting, our findings show that *M. didyma* essential oil is rich in bioactive compounds (mainly aromatic monoterpenes and phenolic monoterpenes) which are most likely responsible for its beneficial effect.

# INTRODUCTION

The exploration of bioactive compounds from natural sources represents an important method for the discovery of new potential therapeutic agents, alternative to chemically synthesized compounds, which often require more complicated and wasteful manufacturing processes. Aromatic plants play a fundamental role in this kind of investigations especially due to their essential oils (EOs), simple to extract and rich in secondary metabolites, with some recognized pharmacological properties. *Monarda*

Corresponding author
Mariastella Colomba,
mariastella.colomba@uniurb.it

**Monarda didyma aerial parts**

**Biological properties**

**Essential oil**

Antioxidant activity

Anti-inflammatory activity

Antimicrobial activity

Antifungal activity

**Figure 1** **Graphical representation of major biological properties of *Monarda didyma* essential oil.**
Many biological properties of *M. didyma* essential oil have been documented in available literature,
the most relevant of which (antifungal/antibacterial, antioxidant and anti-inflammatory activities) are
briefly reported.

*didyma* L. (bergamot or beebalm) (Lamiaceae family) is a perennial herbaceous aromatic
plant native to North America, with verdant coarse leaves and scarlet-red flowers in
terminal tufts (*Carnat, Lamaison & Rémery, 1991*). Recently, *M. didyma* flowers have been
included in the group of edible flowers suitable for human consumption (*Grzeszczuk et al.,
2018*; *Stefaniak & Grzeszczuk, 2019*; *Marchioni et al., 2020*). Moreover, *M. didyma* leaves
and flowers are also used in the preparation of the "Oswego tea" a beverage by the Oswego
tribe of American Indians and imported in Great Britain for traditional English tea. Native
Americans did not use Oswego tea only as a beverage (especially for digestive disorders)
but also for a wide range of medicinal purposes including treatment of fever, headache and
cough, heart ailments, bee stings, skin and mouth (*Fraternale et al., 2006*).

Thanks to its phenols content, this plant has been used for its medicinal properties:
diuretic, antipyretic, sudorific, carminative and antiseptic (*Mattarelli et al., 2017*).
*Monarda didyma* is also characterized by the presence of different bioactive compounds
including carvacrol and thymol that possess beneficial health properties (*Nagoor Meeran
et al., 2017*; *Mahmoodi et al., 2019*; *Javed et al., 2021*).

Thymol and carvacrol have an elevated antioxidant activity for the presence of phenolic
hydroxyls groups (*Beena Kumar & Rawat, 2013*), which suggests that the presence of a
group donor of electrons in the function hydroxylates is crucial to obtain an effective
antioxidant activity.

The chemical composition of aromatic plants is complex and consists of two fractions:
non-volatile and volatile. The last one is composed of secondary metabolites which
constitute the essential oil (EO).

There are a number of studies dealing with isolation and characterization of *M. didyma*
essential oil from flowers or aerial parts (*Mattarelli et al., 2017*; *Wróblewska et al., 2019*;
*Marchioni et al., 2020*; *Côté et al., 2021*) (Fig. 1). First phytochemical studies reported the
presence of flavonoids such as didymin (*Brieskorn & Meister, 1965*; *Scora, 1967*) and linarin (*Ch & Carron, 2007*) from leaves; a bis-malonylated anthocyanin: monardaein (*Saito & Harborne, 1992*) from flowers; and interesting content in thymoquinone and thymohydroquinone from aerial parts, flowers and leaves (*Taborsky et al., 2012*).

The chemical composition of an essential oil (also known as chemotype) may vary considerably for the same species, as the biosynthesis of secondary metabolites is strongly affected by environmental factors, depending either on the geographical origin of the plants or growth conditions (soil, fertility, humidity, sunshine, length of the day) and state of development, which leads to different chemotypes with a specific qualitative and quantitative chemical profiles. Although the species is the same, many differences have been described in chemical composition of *M. didyma* EOs (see for example, *Carnat, Lamaison & Rémery, 1991*; *Fraternale et al., 2006*; *Ricci, Epifano & Fraternale, 2017*). The major compounds representing the chemotype could be thymol (*Shanaida et al., 2021*), linalool (*Carnat, Lamaison & Rémery, 1991*), carvacrol (*Di Vito et al., 2021*), geraniol (*Mazza, Kiehn & Marshall, 1993*) or borneol (*Gwinn et al., 2010*).

The phytochemical analyses on each *M. didyma* essential oil have been associated to *in vitro* or *in vivo* biological and/or physical-chemical studies in order to evaluate a potential application in various fields. In particular, most studies have focused on antifungal/ antibacterial activities, especially to target resistant strains (*Shanaida et al., 2021*) and to search safe alternatives to pure chemical products, such as preservative for food (*Wróblewska et al., 2019*). Moreover, several authors reported about the usage of *M. didyma* essential oil for human health as topical application (*Di Vito et al., 2021*). The antioxidant, antibacterial and anti-inflammatory activities of the essential oil were also reported by physical-chemical analysis or cell-based assay (*Fraternale et al., 2006*; *Côté et al., 2021*).

Among the explored biological properties of *M. didyma* EO, the anti-inflammatory effect deserves additional investigation. The anti-inflammatory activity of essential oils may be attributed not only to their antioxidant properties but also to their interactions with signaling pathway that involves the expression of pro-inflammatory genes and consequently the cytokine production (*Miguel, 2010*). Some authors found that the essential oils studied suppressed the protein and mRNA expression of the cytokines in different lipopolysaccharide (LPS) stimulated cells, assuming that this inhibitory effect seems to be mediated mainly at a transcriptional level (*Gandhi et al., 2020*; *Yadav & Chandra, 2017*; *Yoon et al., 2010*). The stimulation with LPS modulating the Toll-like-receptor-4 (TLR-4) cell signaling pathway can activate inflammatory responses (*Kuzmich et al., 2017*). There are two adapter molecules in this signaling cascade: IL-1 receptor-associated kinase 1 (IRAK-1) and TNF receptor-associated factor 6 (TRAF6); these molecules provoke the activation of NF-kB (nuclear factor-kB), which in turn leads to the production of pro-inflammatory cytokine (IL-6) and miR-146a. Cytokine expression is modulated by TLR-4 through a negative feedback regulation loop involving the down-regulation of IRAK-1 protein level targeted by miR-146a (*Olivieri et al., 2013*). On these bases, as a continuation of a previous study (*Fraternale et al., 2006*), we decided to investigate *M. didyma* essential oil from plants collected in Urbino area (Central Italy, Marche Region) to establish, in addition to its quali-quantitative chemical profile, the

antioxidant activity and to assess the anti-inflammatory effect in an *in vitro* setting. The experimental setting was carried out in order to investigate whether *M. didyma* essential oil anti-inflammatory effect was mediated through TLR-4 signaling pathway and miR-146a negative feedback loop using lipopolysaccharide-stimulated monocyte human tumor cells (U937 cells).

## MATERIALS AND METHODS

### Chemicals and reagents

Thymol methyl ether, carvacrol methyl ether, (+)-2-carene, 1-octen-3-ol, myrcene, trans-anethole, α-terpineol were purchased from Sigma-Aldrich (St. Louis, MO, USA). Carvacrol, γ-terpinene, 3-carene, p-cymene, camphene, α-terpinene, terpinolene, (+/−)-α-pinene, (−/−)-β-pinene, R-(+)-limonene, S-(−)-limonene, were purchased from Extrasynthese (Genay, France). Thymol, linalool, α-phellandrene were purchased from Merck Millipore (Boston, MA, USA). All reagents and solvents have European Pharmacopoeia quality.

### Plant material

The plants were grown in the city of Urbino at 450 m a.s.l. (Marche Region, Central Italy, GPS coordinates: 43°42′50.3″N, 12°36′40.7″E) and identified, by morphological analysis, as *Monarda didyma* L. by Professor Daniele Fraternale from the Department of Biomolecular Sciences (DISB) of the University of Urbino Carlo Bo. The plants were harvested until the beginning of October 2017. The flowering period (September 2017) showed an average maximum temperature of 20.4 °C and an average minimum temperature of 12.6 °C. Precipitation was 162.2 mm and average relative humidity was 74%. These data are available at the "Osservatorio Meteorologico Alessandro Serpieri—Università degli Studi di Urbino Carlo Bo" (https://ossmeteo.uniurb.it/). A specimen of the plant has been preserved in the herbarium of the Botanical Garden of University of Urbino Carlo Bo with accession number: Md 19-63.

### Essential oil isolation

The flowering aerial parts (5.0 Kg fresh weight) of the plants were steam distilled by hydro-distillation method with a yield (v/w) of 2.55 mL/Kg dry weight. The oil was dried and, after filtration, stored at 4 °C until use.

### Essential oil analyses

#### Thin layer chromatography (TLC)

TLC analyses were performed using the following indications of paragraph 2.2.27 'Thin-Layer Chromatography' of the European Pharmacopoeia 7.0 ("European Pharmacopoeia 7.0 Thin-layer chromatography"), and the guidelines of the Technical Guide for the Elaboration of monographs ("EDQM 7 Edition 2015", *EDQM, 2015*). The TLC fingerprint profile was obtained using the following conditions; stationary phase: TLC pre-coated silica gel 60 F254 and HIRSCHMANN® ring caps® 1.2.3.4.5 μL with reproducibility ≤ 0.6% and accuracy ≤ ±0.30% (REF 960 01 05, LOT 861744); essential oil: 0.1 mL/mL in
toluene; reference substances: 6–7 mg/mL; mobile phase: toluene/ethyl acetate (93:7 v/v); spotted volume: 3 µL; start position: 15 mm from the plate edge. The visualization of spots on TLC plates was performed under UV light at 254 nm (UV lamp Benda, NU-8 KL, SN: 6001003): compounds containing at least two conjugated double bonds quench fluorescence and appear as dark zones against the light-green fluorescent background of the TLC plate. We used a vanillin-sulphuric acid solution equal volume of 1% vanillin in anhydrous ethanol w/v and 1% sulphuric acid in anhydrous ethanol v/v as spray reagent and heat for 1 min at 120 °C.

### GC-FID analysis

GC-FID analysis was carried out in an Agilent GC-7820A (Agilent Technologies, Santa Clara, CA, USA) equipped with a Flame Ionization Detector (FID) and coupled to an electronic integrator. The column used was HP-5 column, ref. 19091J-413, lot n° USA563455H (5% Phenyl Methyl Siloxane, 30 m × 0.32 mm i.d. × 0.25 µm); Temperature limits: from −60 °C to 325 °C. The carrier gas was helium (1 mL/min); the injector and detector temperatures were 250 °C and 270 °C, respectively. The samples for the analysis were prepared by diluting 10 µL of essential oil in 1 mL of heptane.

We used two different methods for GC-FID analysis: Method A, the analysis of the *M. didyma* essential oil samples and the reference substances was carried out in sequence for a preliminary identification of the peaks of each compound and to observe their possible presence in the sample by comparing their retention time in the spectra. Method B, the essential oil was co-injected with the various reference substances to confirm their presence in the samples by observing an increase in the intensity of the relative peaks.

In the method A the column temperature was programmed from 40 °C to 220 °C (from 40 °C to 78 °C at a rate of 4 °C/min; from 78 °C to 106 °C at 2 °C/min; from 106 °C to 220 °C at 26 °C/min; then 220 °C for 5 min). Running Time: 32 min 54 s, and 5 min (Post Run) at 40 °C.

In the method B the column temperature was programmed from 40 °C to 300 °C (from 40 °C to 78 °C at a rate of 4 °C/min; from 78 °C to 106 °C at 2 °C/min; from 106 °C to 220 °C at 26 °C/min; from 220 °C to 300 °C at 40 °C/min; then 300 °C for 3 min). Running Time: 34 min 54 s, and 3 min (Post Run) at 40 °C. Compound identification was carried out by comparison of calculated retention indices with those reported in the literature (*Khan et al., 2018*). Two repetitions of essential oil samples at three concentration levels (50, 100 and 150%) were analyzed on three consecutive days.

The quantification of the most abundant compounds was carried out by external calibration from the areas of the chromatographic peaks obtained by GC-FID analysis method A, considering their relative FID areas up to 90% of the total FID area. The choice was based on the verified presence of these compounds by GC-MS analysis, their detection by $^1$H-NMR in CDCl$_3$ solution at 400 MHz, and on the availability of the standards. We did not consider some less abundant compounds with relative peak areas under 2.5% (Table 1). A stock solution of α-terpinene, p-cymene, γ-terpinene, linalool, 1-octen-3-ol, thymol methyl ether, carvacrol methyl ether, thymol and carvacrol was serially diluted with the same solvent to prepare calibration curves ranging from 20–170 mg/mL. The $R^2$
**Table 1 Chemical composition of *Monarda didyma* essential oil (EO) by GC-FID (Method A).** The Retention time of the peaks of the different standards analysed (Ret. Time Standard) and those of the corresponding peaks present in the essential oil spectrum (Ret. Time EO) are indicated. The relative areas (%) of the GC-FID chromatogram peaks are also shown. In bold the most abundant components and their relative areas (>2.5%). The 'Ret. Time EO' represents the mean retention time of two repetitions of essential oil samples (concentration level 100%) analyzed on three consecutive days.

| Compounds | Relative area (%) | Ret. Time EO (min) | Ret. Time Standard (min) |
|---|---|---|---|
| α-Pinene | 0.7 | 10.097 | 10.143 |
| Camphene | 0.3 | 10.607 | 10.621 |
| β-Pinene | — | — | 11.684 |
| **1-Octen-3-ol** | **5.5** | 11.690 | 11.810 |
| β-Myrcene | 2.1 | 12.132 | 12.169 |
| δ-2-Carene | — | — | 12.573 |
| α-Phellandrene | 0.3 | 12.673 | 12.739 |
| δ-3-Carene | 0.1 | 12.916 | — |
| **α-Terpinene** | **2.7** | 13.177 | 13.281 |
| **p-Cymene** | **19.3** | 13.521 | 13.616 |
| D-Limonene | 1.5 | 13.689 | 13.766 |
| Eucalyptol | 0.8 | 13.830 | 13.809 |
| **γ-Terpinene** | **8.8** | 15.026 | 15.071 |
| α-Terpinolene | 0.3 | 16.413 | 16.528 |
| **Linalool** | **2.6** | 16.976 | 17.141 |
| α-Terpineol | 0.5 | 21.773 | 22.006 |
| **Thymol Methyl Ether** | **5.2** | 24.092 | 24.108 |
| **Carvacrol Methyl Ether** | **6.4** | 24.411 | 24.436 |
| **Thymol** | **12.3** | 25.576 | 25.633 |
| **Carvacrol** | **19.5** | 25.753 | 25.754 |

coefficients for the calibration curves were >0.99. We performed two repetitions for each sample at each concentration level (50%, 100% and 150%). All analyses were repeated three times; each time the samples were prepared by the same operator before starting the analysis.

### GC-MS analysis

GC-MS analysis was performed using a Shimadzu gas chromatograph, model GC-MS-QP2010SE, equipped with a quadrupole analyzer ionization mode with electronic impact and DB-5 capillary column (30 m × 0.25 mm i.d. × 0.25 μm, ref. 122–5032, lot n° USR146513H; Agilent Technologies, Santa Clara, CA, USA). The oven temperature was programmed from 40 °C to 220 °C (from 40 °C to 78 °C at a rate of 4 °C/min; from 78 °C to 106 °C at 2 °C/min; from 106 °C to 220 °C at 10° C/min; then 220 °C for 5 min). Running time: 39 min 54 s. Helium was used as carrier gas (constant flow rate 36.1 cm/s). The temperature of ion source and interface were maintained at 220 °C. The injection volume was 1 μL. Prior to injection, the essential oil was diluted (10 μL/1 mL heptane).

The acquisition data and instrument control were performed by the GC-MS Solution software. The identity of each compound was assigned by comparison with the mass spectra characteristic features obtained with the NIST library spectral data bank. For semi-quantification purpose the normalized peak area abundances without correction factors were used. Compounds can be identified by a comparison of their retention index, relative to a standard mixture of n-alkanes (*Adams, 2009*).

### Proton nuclear magnetic resonance ($^1$H-NMR)

The $^1$H-NMR spectra were recorded at 400 MHz for $^1$H experiments on a Bruker Avance 400 MHz spectrometer. NMR Fourier transform, integration and peak picking were done with Bruker Top Spin software. Chemical shifts ($\delta$) are reported in ppm. The one-dimensional spectra were performed in CDCl$_3$. The residual solvent was used as a reference ($\delta$ = 7.26 ppm).

### Determination of Monarda didyma essential oil anti-oxidant activity by DPPH radical assay

The anti-oxidant activity was assessed by DPPH (diphenyl picrylhydrazyl) radical-scavenging method described by *Cuendet et al. (1997)* with slight modifications, as previously described (*Fraternale et al., 2016*). We used 80 µL of 0.5 mM DPPH ethanol solution (Sigma-Aldrich, St. Louis, MI, USA) and 40 µL of *M. didyma* essential oil were diluted in ethanol at final concentrations ranging from 0.10 to 0.50 µL/mL (to achieve a final volume of 400 µL). After 1 h in the dark, the absorbance was measured at 517 nm by UV/Vis spectrophotometer. The control was prepared with 8 µL of 0.5 mM DPPH diluted in 400 µL of ethanol and ethanol without DPPH was used as a blank. Lower absorbance is indicative of higher free radical scavenging activity. The equation to calculate the Inhibitory activity (I) of DPPH radical was the following:

$$I(\%) = 100 \times (A0 - As)/A0$$

In this equation, A0 is the absorbance of the control and As is the absorbance of the tested sample.

Different amounts of *M. didyma* essential oil ethanol solution (0.5 µL/mL) ranging from 20 to 160 µL were used. All the analyses were run in triplicate. EC50 L-ascorbic acid (Sigma-Aldrich, St. Louis, MI, USA) was used as positive control (40 µg/mL final concentration). *M. didyma* essential oil EC50 (50% DPPH scavenging activity) was calculated by regression analysis.

## Biological assays

### Cell culture and treatments

U937 cells (purchased at Euroclone, Milan, Italy) were maintained in RPMI 1640 culture medium (Sigma Aldrich, St. Louis, MI, USA) with 10% fetal bovine serum (FBS), penicillin (50 U/mL) and streptomycin (50 µg/mL) (Euroclone, Milan, Italy). U937 were incubated at 37 °C in tissue culture (T-75 flasks; Corning Inc., Corning, NY, USA) with an atmosphere of 95% air to 5% CO$_2$. To investigate the anti-inflammatory activity of the essential oil, U937 cells were differentiated into macrophages with 100 nM Phorbol 12-

myristate 13-acetate (PMA, Sigma-Aldrich, St. Louis, MI, USA) for 48 h. U937 were then incubated for 24 h without PMA. To expose U937 cells to a pro-inflammatory stimulus, 1 μg/mL lipopolysaccharide (LPS; Sigma-Aldrich, St. Louis, MI, USA) final concentration was added to the medium for 6 h. The essential oil (0.5 μL/mL final concentration) was added 1 h before LPS stimulation. Three replicates were performed.

### Cell viability

To analyze cell viability either Hoechst and Trypan blue have been used to identify apoptosis and necrosis, respectively. Apoptosis was quantified by staining cells with Hoechst 33342 (Sigma-Aldrich, St. Louis, MI, USA). Cells with nuclear apoptotic morphology, detected using a fluorescence microscope (Olympus, Milan, Italy), were counted (at least 100 cells in at least three independent fields) and the fraction of apoptotic cells among total cells was evaluated as percentage (*Radogna et al., 2009*). The percentage of necrotic cells was assessed by trypan blue exclusion test using cell suspension diluted 1:2 (v/v) with 0.4% trypan blue.

### Real time quantitative PCR (RT-qPCR) of mature microRNAs

The total RNA purification kit (Norgen Biotek Corp., Ontario, Canada) was used to isolate total RNA from $1 \times 10^6$ U937 cells, as recommended in the manufacturer's protocol. Samples were subsequently analyzed for nucleic acid quality and quantity using the Nano-Drop ND-1000 spectrophotometer (Nano-Drop Technologies, Wilmington, DE, USA) and stored at −80 °C until use. Human miR-146a and human RNU44 (reference miRNA) expressions were quantified using the TaqMan MicroRNA assay (Applied Biosystems, Foster City, CA, USA) (see *Olivieri et al., 2013*). Experimental protocol was as described in a previous paper by *Fraternale et al. (2016)*. More specifically, the TaqMan MicroRNA reverse transcription kit was used to reverse transcribe the total RNA; 5 μL of RT mix contained 1 μL of each miR-specific has-miR-146a stem-loop primers, 1.67 mL of input RNA, 0.4 μL of 10 × buffer, 0.6 μL of RNAse inhibitor diluted 1:10 and 0.55 μL of $H_2O$. The mixture was incubated at 16 °C for 30 min, at 42 °C for 30 min, and at 85 °C for 5 min. RT-qPCR was performed in 20 μL of PCR mix containing 1 μL of 20 × TaqMan MicroRNA assay—which in turn contained PCR primers and probes (5′-FAM)—10 μL of 2 × TaqMan Universal PCR Master Mix No Amp Erase UNG (Applied Biosystems, Foster City, CA, USA), and 5 μL of reverse-transcribed product. The reaction was maintained at 95 °C for 10 min, then incubated at 40 cycles at 95 °C for 15 s and at 60 °C for 1 min. The RT-qPCR was run on an ABIPRISM 7500 Real Time PCR System (Applied Biosystems, Foster City, CA, USA). Data were analyzed by a 7500-system software (1 1.4.0) with the automatic comparative threshold (Ct) setting for adapting baseline. The relative amount of miR-146a was calculated using the Ct method: $\Delta Ct = Ct$ (miR146a) – Ct (RNU44); $2^{-\Delta Ct}$

Results are expressed as fold change $2^{-\Delta\Delta Ct}$ related to control (CTRL).

### Real time quantitative PCR (RT-qPCR) of mature mRNAs

Total RNA was extracted and analyzed for quality and quantity using the same techniques described before. The RNA extracted was used to synthesize cDNA using a reverse
transcription kit (Applied Biosystems, Foster City, CA, USA) according to the manufacturer's protocol. RT-qPCR was performed with the SYBR Green PCR master mix (Applied Biosystems, Foster City, CA, USA) on an ABI Prism 7500 Real Time PCR System (Applied Biosystems, Foster City, CA, USA). The primers we used were the ones designed by (*Angel-Morales, Noratto & Mertens-Talcott, 2012*). TATA binding protein (TBP) was used as the endogenous reference and forward/reverse primers were purchased from Sigma-Aldrich. Dissociation curve analysis was used to verify product specificity.

Primer sequences were: TBP, forward: TGCACAGGAGCCAAGAGTGAA, reverse: CACATCACAGCTCCCCACCA; IL-6, forward: AGGGCTCTTCGGCAAATGTA, reverse: GAAGGAATGCCCATTAACAACAA; and IRAK-1: forward CAGACAGGGA AGGGAAACATTTT and reverse CATGAAACCTGACTTGCTTCTGAA.

The relative amounts of IL-6 and IRAK-1 were calculated using the Ct method:
$\Delta Ct = Ct$ (IL-6 or IRAK-1) $-$ Ct (TBP); $2^{-\Delta Ct}$
Results are expressed as fold change $2^{-\Delta\Delta Ct}$ related to control (CTRL).

### Statistical analysis

All measurements were expressed as mean values ± standard deviation (SD) from the mean of at least three independent experiments. The two-tailed paired Student's t-test was used for the analyses. The results were considered significant at the level of $p < 0.05$.

## RESULTS

### TLC analysis

TLC analysis showed the presence of thymol, carvacrol, thymol methyl ether, carvacrol methyl ether, 1-octen-3-ol, and linalool. Results obtained for p-cymene, α-terpinene, γ-terpinene, α-terpineol and eucalyptol were uncertain (Fig. 2).

### Qualitative GC-FID analysis

Two different methods (Method A and Method B) were successfully used. GC-FID by method A (analysis, in sequence, of the essential oil samples and the reference substances) resulted in fifteen compounds identified, among which nine were selected as the most abundant ones, whereas the presence of β-pinene, δ-2-carene, δ-3-carene, β-myrcene and α-phellandrene was doubtful (Table 1, see also Fig. S1).

To have further confirmation of the identified compounds, the essential oil was co-injected with the standards (Method B). Compounds identified are listed in Table 2 (see also Fig. S2). Using Method B conditions, the absence of β-pinene and δ-2-carene was confirmed; whereas camphene, δ-3-carene and α-terpinolene showed too small peaks to be identified.

### GC-MS analysis

GC-MS analysis pointed to the presence of different classes of phytochemicals. The identification of each compound was possible thanks to the comparison of mass spectra from the NIST library spectral data bank. We found different compounds with significant values of probability of identification (>95%): α-thujene, α-pinene, camphene, sabinene, β-myrcene, α-phellandrene, δ-3-carene, α-terpinene, p-cymene, γ-terpinene,

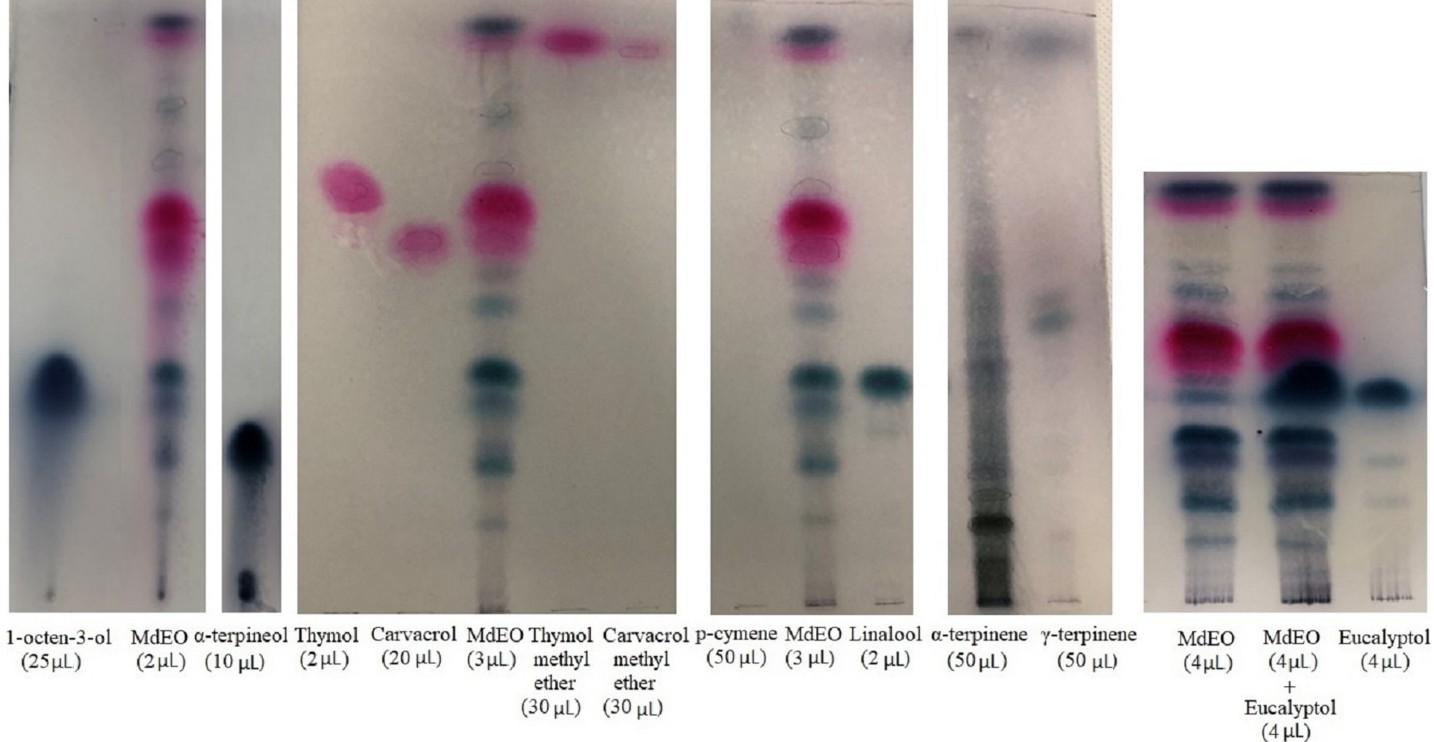

1-octen-3-ol (25 μL)  MdEO (2 μL)  α-terpineol (10 μL)  Thymol (2 μL)  Carvacrol (20 μL)  MdEO (3 μL)  Thymol methyl ether (30 μL)  Carvacrol methyl ether (30 μL)  p-cymene (50 μL)  MdEO (3 μL)  Linalool (2 μL)  α-terpinene (50 μL)  γ-terpinene (50 μL)  MdEO (4 μL)  MdEO (4 μL) + Eucalyptol (4 μL)  Eucalyptol (4 μL)

**Figure 2 TLC of *Monarda didyma* essential oil (EO).** *M. didyma* essential oil (MdEO 2, 3 and 4 μL) analysis with eight standards: 1-octen-3-ol (25 μL), α-terpineol (10 μL), thymol (2 μL), carvacrol (20 μL), thymol methyl ether (30 μL), carvacrol methyl ether (30 μL), p-cymene (50 μL), linalool (2 μL), α-terpinene (50 μL), γ-terpinene (50 μL) and eucalyptol (4 μL).

linalool, α-terpineol, thymol methyl ether, carvacrol methyl ether, thymol and carvacrol; and four with lower percentage: limonene (94%), eucalyptol (94%), β-pinene (91%), α-terpinolene (91%) (Fig. S3).

## ¹H-NMR analysis

*M. didyma* essential oil was also analyzed by ¹H-NMR at 400 MHz in CDCl₃. The main compounds identified were as follows: p-cymene (*Ribeiro, Serra & Rocha Gonsalves, 2010*), carvacrol (*Han & Armstrong, 2005*), thymol (*Chung et al., 2007*), carvacrol methyl ether (*Narkhede et al., 2008*), 1-octen-3-ol (*Felluga et al., 2007*), thymol methyl ether (*Maraš, Polanc & Kočevar, 2008*), linalool (*Shibuya, Tomizawa & Iwabuchi, 2008*), γ-terpinene (*Ishifune et al., 2003*) and α-terpinene (*Utenkova et al., 2017*) (Fig. 3); due to lower concentration in the mixture, and to signal overlapping, α-pinene, α-terpineol, limonene, myrcene, and camphene were difficult to detect.

In particular, despite the expected overlap of most of the resonance signals, some portions of the spectra can be decoded, and some set of signals can be assigned to single compounds, already identified by GC-FID/GC-MS analysis. Most significant resonances in the ¹H-NMR spectrum reported in Fig. 3, were as follows: a multiplet at 7.15 ppm for the aromatic protons of p-cymene; a series of overlapped doublets in the 6.7–7.1 ppm range for the aromatic H5 and H6 of carvacrol, thymol, carvacrol methyl ether and thymol methyl ether; two broad singlets at 6.68 and about 6.6 ppm for the aromatic H3 of carvacrol and

**Table 2 Chemical composition of *Monarda didyma* essential oil by GC-FID (Method B).** Compounds identified by GC-FID in the essential oil co-injected with the standards. The retention time of the peaks of the essential oil (Ret. Time EO) and that of the essential oil co-injected with the standards (Ret. Time Standard co-injected with the EO) are indicated.

| Compounds | Ret. Time EO (min) | Ret. Time Standard co-injected with the EO (min) |
|---|---|---|
| α-Pinene | 10.142 | 10.131 |
| Camphene | — | 10.640 |
| β-Pinene | — | 11.658 |
| 1-Octen-3-ol | 11.746 | 11.723 |
| β-Myrcene | 12.024 | 12.169 |
| δ-2-Carene | — | 12.577 |
| δ-3-Carene | — | 12.954 |
| α-Terpinene | 13.230 | 13.211 |
| p-Cymene | 13.571 | 13.563 |
| D-Limonene | 13.743 | 13.739 |
| γ-Terpinene | 15.083 | 15.066 |
| α-Terpinolene | — | 16.447 |
| Linalool | 17.048 | 17.037 |
| Thymol Methyl Ether | 24.140 | 24.141 |
| Carvacrol Methyl Ether | 24.453 | 24.446 |
| Thymol | 25.604 | 25.595 |
| Carvacrol | 25.777 | 25.791 |

carvacrol methyl ether, and the aromatic H2 of thymol and thymol methyl ether; two overlapped multiplets at 5.9 ppm for the olefinic H2 (proximal to the -OH substituent) of both 1-octen-3-ol and linalool; an isolated broad multiplet in the 5.6–5.7 ppm range for the olefinic H2 and H3 of α-terpinene; a broad singlet at 5.47 ppm for the olefinic H2 and H5 of γ-terpinene; a set of overlapped doublets from 5 to 5.3 ppm for the olefinic H1 of 1-octen-3-ol and olefinic H1 and H6 of linalool; a broad singlet for both the OH protons of thymol and carvacrol; a broad isolated quartet at approximately 4.1–4.2 ppm for the allylic H3 of 1-octen-3-ol; two isolated and distinctive singlets for the methoxyl substituent of carvacrol methyl ether (OMe2) and thymol methyl ether (OMe3), at 3.87 and 3.84 ppm respectively; a set of multiplets typical for the isopropyl C-H (H7), at 3.3 ppm for thymol methyl ether, at 3.2 ppm for thymol, and from 2.8 to 3 ppm for carvacrol, carvacrol methyl ether, p-cymene and α-terpinene; a broad signal at approximately 2.6–2.7 ppm, for methylene protons H3 and H6 of γ-terpinene; a set of five distinct singlets in the 2.2–2.4 ppm range, for the methyl substituent on the aromatic ring (Me1) of carvacrol methyl ether, carvacrol, thymol, p-cymene and thymol methyl ether; a series of overlapped doublets at 1.2–1.3 ppm, for the geminal methyl protons of the isopropyl substituent (Me7) of thymol methyl ether, p-cymene, thymol, carvacrol, carvacrol methyl ether, and two overlapped doublets at 1.0–1.1 ppm, for the same type of protons (Me7) of γ-terpinene and α-terpinene.

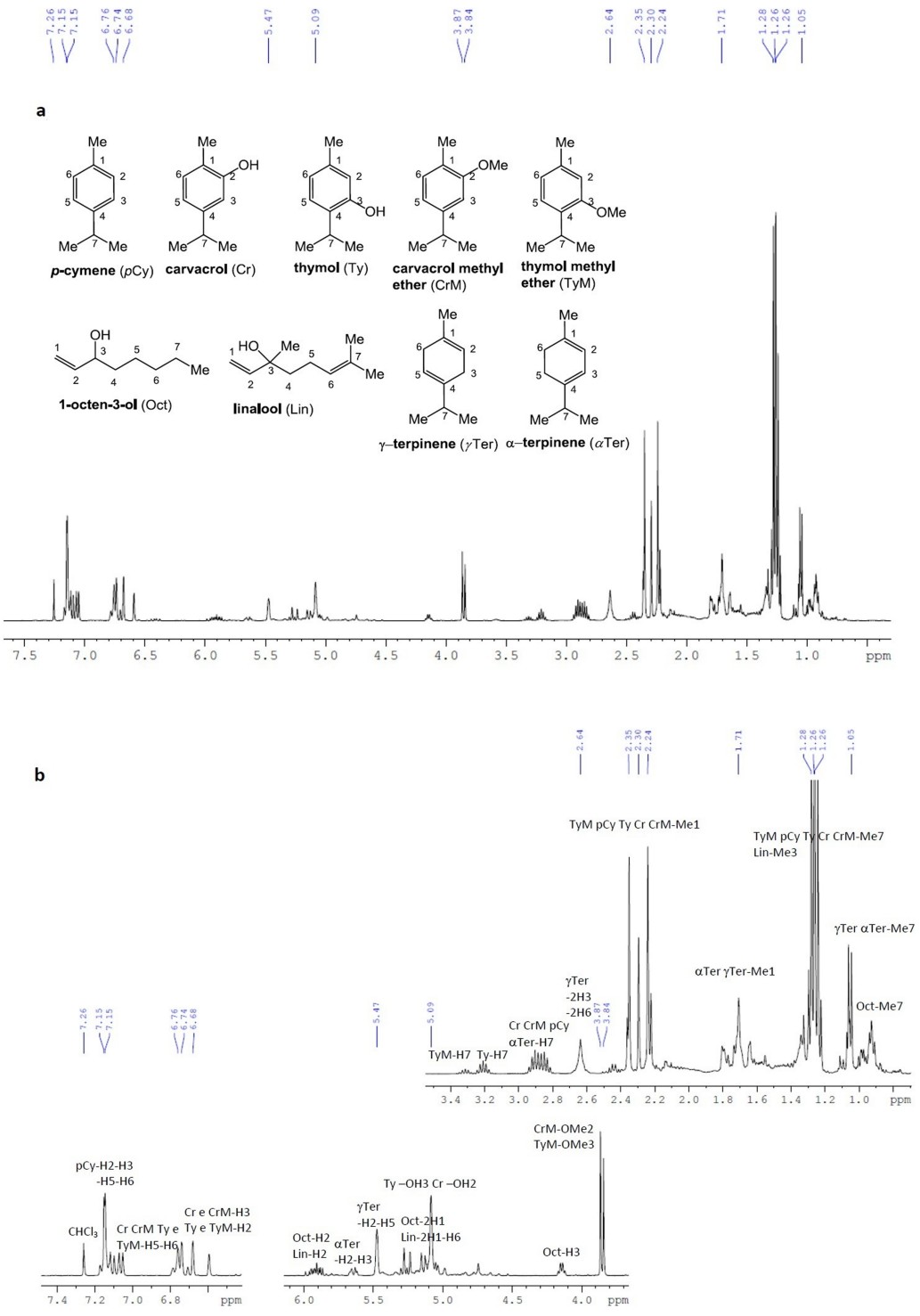

**Figure 3 ¹H-NMR spectra of *Monarda didyma* essential oil at 400 MHz in CDCl₃.** (A) ¹H-NMR spectra of *Monarda didyma* essential oil at 400 MHz in CDCl₃ (δ 7.26 ppm) and structures with suitable numbering of the detectable compounds; (B) enlarged ¹H-NMR fragments with annotation for the assignment of the most significant chemical shift ranges or peak resonances. The main constituents detectable in the essential oil were the following: p-cymene (pCy), carvacrol (Cr), thymol (Ty), carvacrol methyl ether (CrM), 1-octen-3-ol (Oct), thymol methyl ether (TyM), linalool (Lin), γ-terpinen (γTer) and α-terpinen (αTer).

**Table 3 Quantitative GC-FID analysis of *Monarda didyma* essential oil (EO).** Quantities of the most abundant compounds (ca. 72% of the entire essential oil composition), expressed as mg/mL and percentage ± standard deviation, are listed below.

| Compounds | mg/mL EO | % (g/100 mL) ± SD |
|---|---|---|
| Carvacrol | 167.439 | 16.744 ± 0.318 |
| p-Cymene | 159.999 | 16.000 ± 0.972 |
| Thymol | 105.135 | 10.513 ± 0.195 |
| γ-Terpinene | 78.095 | 7.810 ± 0.384 |
| Carvacrol methyl ether | 56.725 | 5.673 ± 0.204 |
| 1-Octen-3-ol | 53.700 | 5.370 ± 0.001 |
| Thymol methyl ether | 46.270 | 4.627 ± 0.138 |
| Linalool | 25.212 | 2.521 ± 0.068 |
| α-Terpinene | 24.336 | 2.434 ± 0.078 |
| | TOTAL % | 71.691 |

## Quantitative GC-FID analyses

The concentrations ± standard deviation (SD) of the most abundant compounds in *M. didyma* essential oil are indicated in Table 3.

## Essential oil antioxidant and anti-inflammatory activity

The essential oil showed a 50% DPPH free radical scavenging activity with 160.214 μL of *M. didyma* essential oil ethanol solution (0.5 μL/mL), corresponding to the effect exerted by 0.4 μL of Ascorbic Acid ethanol solution (40 mg/mL) in a final volume of 400 μL. The analysis demonstrated a dose-dependent effect ($R^2$ = 0.9803) (Fig. 4A).

The anti-inflammatory effect of *M. didyma* essential oil was investigated treating U937 cells with an essential oil ethanol solution (0.5 μL/mL final concentration) during a pro-inflammatory stimulus (LPS, 1 μg/mL final concentration). We used this dose of essential oil since we found that it had a good antioxidant activity without creating any cellular toxicity (Fig. 4B). LPS treated U937 cells showed the typical inflammatory condition: down regulation of miR-146a expression level (Fig. 5A) and high amounts of the pro-inflammatory markers IRAK-1 and IL-6 (Figs. 5B and 5C). When treating the cells with the essential oil (MdEO), we observed that the anti-inflammatory effect was particularly evident, resulting in an overexpression of miR-146a (Fig. 5A) and the consequent down regulation of IRAK-1 and IL-6 (Figs. 5B and 5C). When considering MdEO-pretreated U937 cells, under a pro-inflammatory stimulation (LPS + MdEO), the up-regulation of miR-146a was evident as well but, as expected, a little lower than that observed in MdEO experimental group (Fig. 5A), thus confirming that the essential oil phytochemical compounds can efficiently cope with the inflammatory cascade triggered by the LPS insult. The ability of *M. didyma* essential oil to modulate the inflammatory response, through the inhibition of the TLR-4 signaling pathway and the reduced expression of IL-6 (Fig. 5D), demonstrates that it is a down modulator of transcriptional regulation of pro-inflammatory molecules, at least in an *in vitro* setting. This result is suggestive of a potential good *in vivo* anti-inflammatory activity.

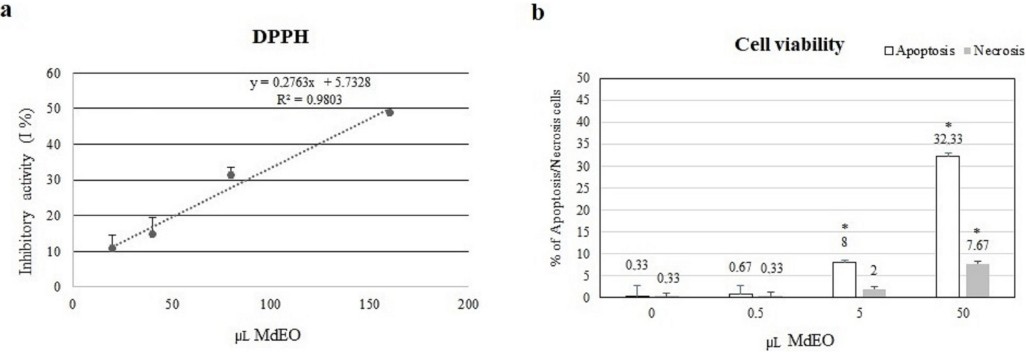

**Figure 4 DPPH antioxidant activity and cell viability.** (A) The essential oil showed a 50% DPPH scavenging activity with 160.214 μL of *M. didyma* essential oil (MdEO) and the analysis demonstrated a dose-dependent effect ($R^2 = 0.9803$). (B) Cell viability (apoptosis and necrosis) has been evaluated in U937 cells treated at different concentrations of *M. didyma* essential oil (MdEO): 0, 0.5, 5 and 50 μL. The values are expressed as mean values ± standard deviation (SD) from the mean of at least three independent experiments. The two-tailed paired Student's t-test was used to compare the results *vs* 0 μL of MdEO (*$p < 0.05$).

## DISCUSSION

Essential oils (EOs) from aromatic and medicinal plants are known to possess biological activity. In order to provide our contribute in investigating plant-derived phytochemical compounds we studied the chemical composition of the essential oil of *M. didyma* aerial parts from Urbino (Central Italy, Marche region) and evaluated the antioxidant activity and its potential as an anti-inflammatory agent in an *in vitro* setting. According to the quali-quantitative chemical analysis, twenty compounds were identified. Among these, we quantified the most abundant ones including carvacrol, p-cymene, thymol, γ-terpinene, carvacrol methyl ether, 1-octen-3-ol, thymol methyl ether, linalool and α-terpinene.

Taking into account that the chemical profile of the essential oils can be greatly variable, especially due to genetic or climatic causes but also to geographic origin, we compared the chemical composition of the *M. didyma* essential oil examined in this experimental set (*i.e.*, *M. didyma* ct. carvacrol) with those provided for two Italian chemotypes (*Fraternale et al., 2006*; *Ricci, Epifano & Fraternale, 2017*) and one French chemotype (*Carnat, Lamaison & Rémery, 1991*).

Several compounds that we identified correspond to those previously reported by the authors cited above, such as α-pinene, camphene, β-myrcene, p-cymene, limonene, linalool and α-terpineol, but, as expected, they show different concentration profiles, according to the chemotype studied. On the other hand, compounds such as carvacrol and carvacrol methyl ether, that we found in concentration of 16.74% and 5.67% respectively, have not been mentioned in those previous studies.

We found remarkable differences just comparing the composition of the essential oils from two *M. didyma* plants of related origin, from Italy: one cultivated in Urbino (same cultivation area of our plant) (*Fraternale et al., 2006*) and the other one in Imola (*Ricci, Epifano & Fraternale, 2017*).

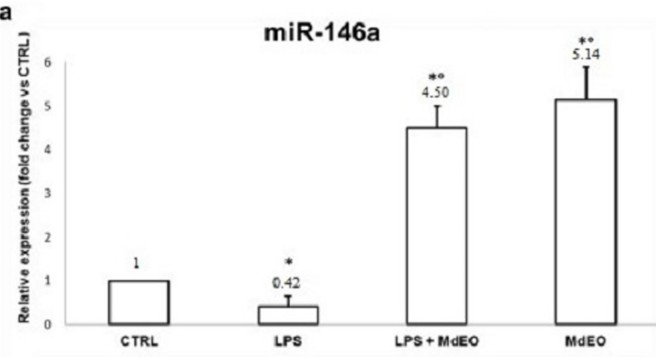

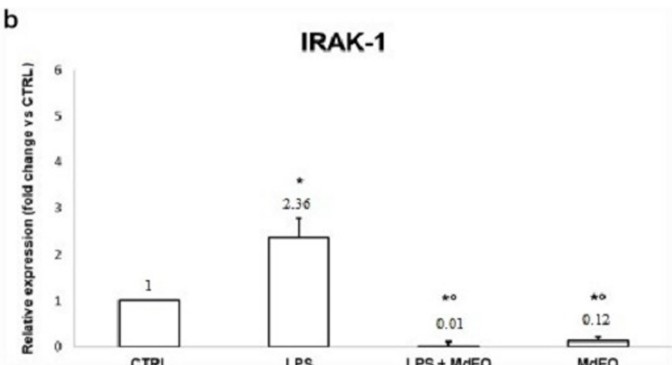

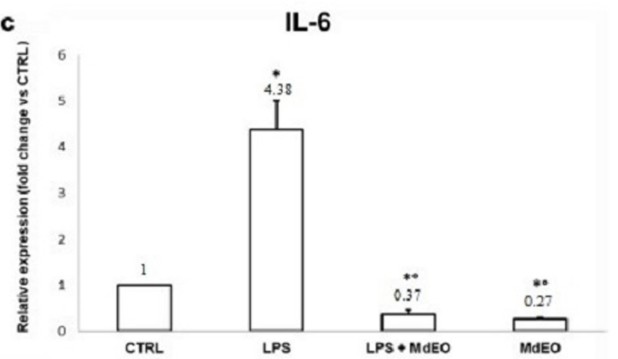

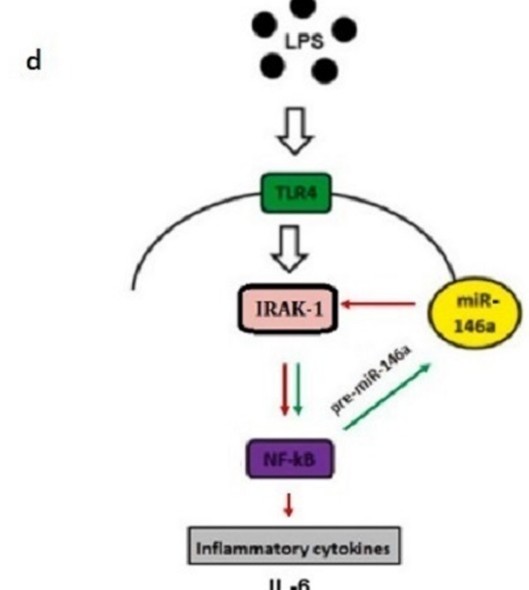

**Figure 5 Anti-inflammatory effect of *Monarda didyma* essential oil on miR-146a, IRAK-1 and IL-6 expression levels in U937 cells.** Both experimental groups treated with the essential oil (LPS + MdEO and MdEO,) showed miR-146a up-regulation (A) with a decreased expression of IRAK-1 and IL-6 (B and C). Results are reported as fold change related to CTRL. CTRL: controls; LPS: LPS-stimulated cells (1 µg/mL final concentration); LPS + MdEO: cells treated with MdEO (0.5 µL/mL final concentration) 1 h before LPS stimulation (1 µg/mL final concentration); MdEO: cells treated with the essential oil (0.5 µL/mL final concentration). Two-tailed paired Student's t-test: $^*p < 0.05$ *vs* CTRL; $^{\circ}p < 0.05$ *vs* LPS. TLR-4 signaling pathway is illustrated in panel D. Red arrow: negative interaction; Green arrow: positive interaction.

In particular, in these other Italian *M. didyma* EOs, the most abundant compounds were thymol (51.7–59.3%) and p-cymene (10.5–9.7%), while in our essential oil the most abundant compounds are carvacrol (16.74%), p-cymene (16.00%) and thymol (10.51%).

Considering the composition of the two essential oils obtained from plants grown in Urbino (*Fraternale et al., 2006* and present paper), the observed differences could be ascribed to the different season (solar inclination, luminosity, temperature) and different climatic conditions that occurred in the two distinct flowering periods; July 2006 and

September 2017. In July 2006, temperatures on average were higher (maximum temperature: 27.1 °C *vs* 20.4 °C, minimum temperature: 18.2 °C *vs* 12.6 °C); it was less rainy (precipitation: 35 mm *vs* 162.2 mm) and a little less humid (relative humidity: 55% *vs* 74%) (https://ossmeteo.uniurb.it/).

As concerns the most abundant compounds that are most biologically interesting in our essential oil, there is an extensive literature reporting on the pharmacological activities and therapeutic potential of thymol in view of its antioxidant, anti-inflammatory and anti-tumoral properties (*Nagoor Meeran et al., 2017*; and references therein). In particular, the antioxidant and anti-inflammatory properties of thymol have been well documented in various preclinical studies including cell lines and animal models (see for example, *Vigo et al., 2004*; *Marsik et al., 2005*; *Braga et al., 2006*; *Undeger et al., 2009*; *Archana, Nageshwar Rao & Satish Rao, 2011*; *Nagoor Meeran & Prince, 2012*; *Chauhan et al., 2014*; *Cabello et al., 2015*; *Nagoor Meeran, Jagadeesh & Selvaraj, 2015, 2016*; *Perez-Roses et al., 2016*; *Zidan et al., 2016*; *Wei et al., 2017*).

As far as concerns carvacrol, a number of research studies have shown biological actions of carvacrol as an immunomodulator agent (*Mahmoodi et al., 2019*) with a great therapeutic potential. The *in vitro* and *in vivo* studies have shown multiple pharmacological properties such as anticancer, antifungal, antibacterial, antioxidant, anti-inflammatory, vasorelaxant, hepatoprotective, spasmolytic, immunomodulating and anti-viral (*Javed et al., 2021*).

p-cymene is an aromatic monoterpene with a widespread range of therapeutic properties including antioxidant and anti-inflammatory activity (see *de Oliveira Formiga et al., 2020*; *Sani et al., 2022*).

Finally, carvacrol methyl ether has been shown to have antibacterial activity with a high potential in the food industry and agriculture (*Simirgiotis et al., 2020*).

Recent studies have demonstrated that flavonoids (*Maleki, Crespo & Cabanillas, 2019*) and phenolic compounds (*Yahfoufi et al., 2018*) can inhibit regulatory enzymes or transcription factors important for controlling mediators involved in inflammation.

Our findings indicate that during an acute LPS stimulation in *M. didyma* essential oil pretreated U937 cells (MdEO + LPS), miR-146a increases (Fig. 5A) and targets IRAK-1, decreasing its expression (Fig. 5B), with a consequent reduction of IL-6 amounts (Fig. 5C). Such a result strongly supports the hypothesis that the mechanism of action by which *M. didyma* essential oil exerts its anti-inflammatory activity involves the TLR-4 signaling pathway (Fig. 5D).

Based on the wide range of experimental studies reporting the potential of thymol, carvacrol and p-cymene as antioxidant and anti-inflammatory drugs (see the above-mentioned literature), we hypothesize that the antioxidant property of the essential oil of *M. didyma* ct. carvacrol we observed in this study is probably due to one or more of these phytochemicals which may act by scavenging free radicals and prevent lipid peroxidation. Moreover, one of the three most abundant compounds alone, or a synergistic interaction of a particular combination of the secondary EO metabolites, may be primarily responsible for the observed anti-inflammatory effect, thus making *M. didyma* ct. carvacrol suitable candidate for further investigation.

## CONCLUSIONS

Our study highlights an aspect still unexplored by the scientific literature regarding the real anti-inflammatory activity of *M. didyma* essential oil. Although further studies are needed for a better elucidation of the molecular mechanisms underlying the observed associated antioxidant and anti-inflammatory bioactivities of *M. didyma* essential oil examined in the present study, they are most likely due to the bio-functional properties of the monoterpenes (mainly aromatic and phenolic monoterpenes) present in the essential oil. As far as concerns the anti-inflammatory activity, in our experimental set, LPS-stimulated U937 cells showed a decreased expression of IRAK-1 and IL-6 (a pro-inflammatory cytokine) and an increased expression of miR-146a, which are suggestive of the involvement of the Toll-like receptor-4 signaling pathway.

In summary, this study characterizes a new special chemotype of *M. didyma* essential oil and expands the knowledge about its biological activity confirming that it may be considered, after normalization, in herbal medicinal products or food supplements and could be proposed as a natural source of bioactive compounds.

## ACKNOWLEDGEMENTS

The authors would like to express their special thanks to the referee Kristine L. Trotta whose suggestions and comments contributed to greatly improve the manuscript. The authors also wish to express their gratitude to the other three (anonymous) referees for their insights.

### Funding

The present project was made possible thanks to the ERASMUS+ Mobility for Studies programme which awarded Rossella D'Adderio a scholarship and enabled her to spend a period of continuous study at the Université de Paris, Faculté de Santé, Pharmacie that has entered into an agreement with the University of Urbino Carlo Bo. The funders had no role in study design, data collection and analysis, decision to publish, or preparation of the manuscript.

### Grant Disclosures

The following grant information was disclosed by the authors:
Université de Paris.
University of Urbino Carlo Bo.

### Competing Interests

Maria Cristina Albertini is an Academic Editor for PeerJ. The authors declare that they have no additional competing interests.

## Author Contributions

- Daniele Fraternale performed the experiments, authored or reviewed drafts of the article, and approved the final draft.
- Hanh Dufat conceived and designed the experiments, authored or reviewed drafts of the article, and approved the final draft.
- Maria Cristina Albertini conceived and designed the experiments, authored or reviewed drafts of the article, and approved the final draft.
- Chouaha Bouzidi performed the experiments, prepared figures and/or tables, and approved the final draft.
- Rossella D'Adderio performed the experiments, analyzed the data, prepared figures and/or tables, and approved the final draft.
- Sofia Coppari performed the experiments, analyzed the data, prepared figures and/or tables, and approved the final draft.
- Barbara Di Giacomo analyzed the data, authored or reviewed drafts of the article, and approved the final draft.
- Davide Melandri analyzed the data, prepared figures and/or tables, and approved the final draft.
- Seeram Ramakrishna analyzed the data, prepared figures and/or tables, and approved the final draft.
- Mariastella Colomba conceived and designed the experiments, analyzed the data, authored or reviewed drafts of the article, and approved the final draft.

## Data Availability

The GC-FID Chromatogram (Method A), GC-FID Chromatogram (Method B) and GC-MS Chromatogram are available in the Supplementary Files.

## Supplemental Information

Supplemental information for this article can be found online at http://dx.doi.org/10.7717/peerj.14433#supplemental-information.

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
