# Peer review of "Chemical composition, antioxidant and anti-inflammatory properties of Monarda didyma L. essential oil"

_PeerJ, doi:10.7717/peerj.14433_

## Round 0.1 · original submission · Major Revisions

Please, look carefully at all comments of the referees.

Reviewer 1 ·

Basic reporting

English language in this manuscript is OK. Some typos can be found in following comments. However, writing of this manuscript is not professional, especially in citation formats and data presentation. I recommend authors find a colleague proficient in English scientific writing to review this manuscript.
Citation and reference bibliography formats need corrections.
The manuscript has professional article structure. Some repetitions might be included in figures/tables.

Experimental design

This manuscript is within the aims and scope of PeerJ.
Research quesion is defined well.
More details needed in Materials and methods, please see following comments.

Validity of the findings

Meaningful replication need to be presented.
Conclusion need corrections since some part in conclusion should be included in discussion.

Additional comments

Major revision:
Origin information about the plants used in this study need to be introduced to make this study replicable.
Result of TLC analysis: Do you have any picture on the TLC analysis? You may attach them to the manuscript or the supplementary materials.
There is no standard deviation in Table 3, how many repetitions were tested? Except for GC-FID, the repetition time for different chemical analyses were not mentioned. Data presentation in this manuscript is not professional.
L325-326: Since many studies tested chemical compositions in essential oil of M. didyma, and the results are very diversified depending on the parameters mentioned here. There should be a discussion on the chemical composition and content comparation based on results in this study and previous studies.
L52, 64-66: Citation formats need to be corrected. Please carefully check similar problems in the whole manuscript. For example, L469, there is citation information in the link as “Mazza, G., F.A. Kiehn, and H.H. Marshall. 1993. Monarda: A source of geraniol, linalool, thymol and carvacrol-rich essential oils. p. 628-631. In: J. Janick and J.E. Simon (eds.), New crops. Wiley, New York.”
L83: Since quantitative chemical profile analysis is the objective of study, more content data should be included, but not just 9 ‘major’ chemical contents without standard deviations.
Minor revision:
Longitude and latitude, temperature, precipitation, and climate in Urbino need a brief introduction.
L99: The plants were identified by morphology or genetic information?
L245: The format of the citation is not right. Please correct it.
Table 1: Does the ‘Ret. Time EO’ represent mean retention time of different repetitions? If yes, you may explain this in legend.
L297: Please italicize the scientific name.
Table 3: Please explain EO in the legend like what you did in Table 1 & 2.
L297: Why concentration is shorted as Z? I didn’t see Z was used in Table 3.
L364-367: This part might be included in the discussion. However, it should not be a part of conclusion.

Reviewer 2 ·

Basic reporting

The manuscript is well written, english is correct throughout. The structure is good and data presented support the hypotheses. Figures and tables layout is good and easy to read.

Experimental design

The topic and findings of the manuscript fall within the Aims and Scopes of the journal. the research work is well panned and the layout of experimental design is clear and sufficiently detailed.

Validity of the findings

the manuscript investigate the composition and biological effects of an italian variety of Monarda didyma L. essential oil, growing in the Urbino region. Although some previous result on the same matrice is available in the literature, this specific variety is quite unreported for. results of the antioxidant and antinflammatory effects are interesting and although leaving open questions regarding the mechanism of action, are statistically sound and represent an excellent starting point for further investigation.

Additional comments

I think this paper is scientifically relevant and should be published as it is.

Reviewer 3 ·

Basic reporting

This article is well written with clear and unambiguous professional English. All materials are clearly laid out. Figures are relevant, well labeled and described. Related studies are well referenced in the introduction and background. I have a few minor suggestions for quality improvement as follows: 1) to show the actual image of Monarda didyma L. used in this study instead of graphical representation for figure 1. 2) to include a diagram of the Toll-like receptor-4 signaling pathway in figure 3 to help readers who are unfamiliar with this signaling pathway to better understand the interaction between miR-146a, IRAK-1 and IL-6 investigated in this study. 3) To compare and contrast the results of chemical composition by different methods and combine those results into one section.

Experimental design

This article is a study on the chemical composition, antioxidant and anti-inflammatory activity of the essential oil of Monarda didyma L. aerial parts from Urbino (Central Italy, Marche region). There are multiple existing studies on the chemical composition of Monarda didyma L., but the investigation of anti-inflammatory effects is currently unavailable. Therefore, this report has filled in the gap for this field. Appropriate methods were performed to collectively identify the chemical composition including LC, GC-FID, GC-MS and 1H-NMR. The in vitro anti-inflammatory activity of the essential oil was examined by a proper and relevant model study of LPS stimulated cells. All methods are described with sufficient details and information to replicate.

Validity of the findings

Provided data are presented in an appropriate fashion. They are robust, statistically sound, and well controlled and support the conclusions. I suggest that the authors also include the data to support a dose-dependent effect of essential oil on antioxidant activity and non cellular toxicity of selected dose performed in the anti-inflammatory activity study.

·

Basic reporting

1. For the most part, the writing is comprehensible and the grammar is correct. However, there are some instances where the English language can be improved to provide clarity to the reader. For example, there are some convoluted sentences that are hard to understand and could greatly benefit from being edited for simplicity (e.g., lines 54-59, 74-77, 358-360, 364-369). Also, there are some instances of incorrect pluralization of certain words: line 77 “cytokines” should be “cytokine”; line 147 “compounds” should be “compound”. I can understand how some of these grammatical inconsistencies are quite nuanced. Thus, I would recommend that the authors ask another colleague who writes in English (and is familiar with the subject material) to review the manuscript.

2. In lines 45 and 52, the authors cite a Wikipedia article. According to Wikipedia itself, its articles are not a reliable source for academic research purposes (https://en.wikipedia.org/wiki/Wikipedia:Academic_use). Please update the introduction and citations to include peer-reviewed articles for the background on M.didyma.

3. Figures 1S and 2S contain images for GC spectra that are of poor resolution. The numbers are barely legible, making them difficult to interpret. I strongly recommend that the authors resubmit higher-resolution versions of these images. In addition, I recommend that the authors provide supplemental figure legends for 1S and 2S to help orient the reader’s interpretation and their relevance to the text.

4. I commend the authors for the visual clarity of the qPCR plots in Figure 3. The results were clear, and mostly easy to interpret. I have a few suggestions to improve the clarity of the data visualization and integration into the main text.
a. Figure 3 contains panels “a”, “b”, and “c”, but the panels are not individually cited in the text or figure legend. Please update the figure legend to individually describe the data represented in panels a-c. Additionally, please include references to the individual panels in the Results and Discussion sections to aid in the specificity of the text.
b. I recommend modifying the Y-axes to Log2 or Log10 to capture the range of fold-changes between 0 and 1. Right now, all fold-changes <1 are difficult to interpret because they are so close to 0 in the linear Y-axis scale. Alternatively, the authors could instead print the fold-change above each bar to help the reader interpret the fold-changes numerically.
c. Please increase the font sizes for the statistical test symbols “°” and “*” to improve their readability.

5. I strongly recommend that the authors update and annotate Figure 2 (1H NMR) to help integrate the data with the text.
a. Please annotate the solvent peak (CDCl3).
b. In line 293, the authors state “1H-NMR spectra at 300 MHz in CDCl3 of standard compounds were used for comparison”, but these spectra are not included in the manuscript and/or are not cited. Furthermore, Figure 2 is not annotated, so it is impossible to know if the compounds described as being identified by 1H NMR are present in the sample. Please include or cite the reference spectra, and update annotations as appropriate.

Experimental design

1. As currently written, it is not obvious how this study fills a clear knowledge gap. This study is similar in spirit to previous works (e.g., Fraternale et al. 2006, Côté et al. 2021), which also define the chemotype of M. didyma essential oils and their in vitro bioactivity. A considerable portion of the introduction is dedicated to describing these previous works, but the authors do not clearly rationalize a need to perform this study. This is particularly confusing because the cited previous works use similar methods to extract the essential oil, analyze its chemotype, and test its immunomodulatory or anti-inflammatory activity. With this context in mind, the authors would benefit from strengthening their introduction to explicitly tell the reader how this study can be distinguished from previous works in the same field. I believe this can be achieved by choosing stronger language at the end of the introduction (lines 81-86) which clarifies the scientific rationale for generating this specific M.didyma essential oil chemotype and in testing its antioxidant and anti-inflammatory activities.

2. The methods are generally well-described and contain enough detail to replicate. I commend the authors for the clarity, specificity, and use of citations in their methods section.

3. The authors report nine chemical compounds that make up the majority of the Urbino M.didyma chemotype, as determined by GC-FID analysis (line 149, 363). Relatedly, they state that they did not include “some minor compounds” (line 156) in their analysis. As written, it is not clear how they discriminated between “major” and “minor” compounds. Can the authors include an explanation in the text that explains how they used their data to decide which chemical compounds were “major” or “minor”? For example, it would be helpful to know if the authors assigned a cut-off for value for relative abundance, or cross-validated the compound’s presence with their other analyses, or used other methods not listed.

Validity of the findings

1. In the introduction and discussion, the authors call out several plant-derived chemical compounds which are of interest for their bioactivity (lines 52-59,330-332, 337-8). They also hypothesize that the monoterpenes identified in this study may be the source of the antioxidant and anti-inflammatory activities in the Urbino M.didyma essential oil (lines 364-367). I appreciate this line of thinking, as it connects their findings to an exciting hypothesis about the molecular mechanisms of the essential oil’s bioactivity. To further strengthen this line of discussion, I recommend that the authors comment on which specific chemical compounds in their M.didyma essential oil may be strong candidates to explain its antioxidant and anti-inflammatory activities. For example, thymol is a major compound in the Urbino chemotype (line 338) and has been shown to have both anti-inflammatory and antioxidant properties in numerous laboratory studies (see reviews like DOI: 10.3389/fphar.2017.00380). Integration of a discussion about the bioactivities of major compounds in the Urbino chemotype (with appropriate citations) can significantly strengthen the paper’s justification for the utility of the chemotype, and can be a great source for hypothesis generation for readers and future scientists.

2. In the discussion, the authors make important comparisons between the Urbino chemotype, and other M.didyma chemotypes. They identify many compounds that are common across M.didyma essential oils (lines 331-2), but that there are often differences in relative abundance between the published chemotypes. Also, they identify the compounds carvacrol and carvacrol methyl ether as novel components of the Urbino chemotype, which were not found in other studies (lines 333-4). As a reader, I would appreciate if the authors could expand their discussion of these chemotype-specific differences. For example, how might the differences in relative compound abundance affect the antioxidant activity across M.didyma samples? How might the relative abundance of a given compound in the Urbino chemotype impact the plant’s utility for pharmaceutical or industrial applications? An expanded discussion of the practical implications of this chemotype would help strengthen the manuscript by connecting the data to the broader context of this plant’s apparent industrial utility.

3. The authors report performing a thin-layer chromatography analysis of the M.didyma essential oil (lines 110-124, 264-267), and claim several chemical compounds to be present based on the data. However, these data are missing from the paper. Please include images of the TLC plates with appropriate controls (e.g., reference compounds as defined in the methods) and annotations.

4. In the methods, the authors report using two types of NMR instruments (300 MHz and 400 MHz) to take 1H NMR spectra of the Urbino M.didyma essential oil and reference spectra. However, the only data included in the manuscript is a 400 MHz spectrum (Fig 2). Please update the 1H NMR methods to reflect the data included in the manuscript.

5. In lines 307-308, the authors describe an assay where they treat U937 cells with the M.didyma essential oil. They justify the dose of essential oil used by writing “we used this dose of essential oil since we found that it had a good antioxidant activity without creating any cellular toxicity (data not shown)”. In writing this, the authors imply that M.didyma essential oil can be toxic to U937 cells in this assay, but do not show data to justify that the working concentration of essential oil used is not toxic. Please include the toxicity data for M.didyma essential oil against U937 cells to justify the final working concentration used.

6. In lines 317-320, the authors claim, “The ability of M. didyma essential oil to modulate the inflammatory response, through the inhibition of the TLR-4 signaling pathway and the reduced expression of IL-6, demonstrates that this is a good anti-inflammatory agent”. I believe this statement overstates the interpretation of the qRT-PCR data. In essence, the qRT-PCR data show that M.didyma essential oil modulates the basal transcription of miR-146a, IRAK-1, and IL-6, such that the TLR4 signaling pathway appears to lose sensitivity to LPS stimulation. These data are suggestive of the anti-inflammatory activity of M.didyma in an in vitro setting, but qRT-PCR data alone are not sufficient to make a claim about if the anti-inflammatory activity is “good”. I recommend the authors update the interpretation of this data to be more specific to the scope of the assay (transcriptional regulation).

7. Lines 343-351 contain background information about the TRL4 signaling pathway. This information is important for justifying the target transcripts chosen for the qRT-PCR analysis, but it does not contribute meaningfully to the discussion session. I would recommend moving it into the introduction or methods, as rationale for the study design.

Additional comments

This study concerns the chemical composition and bioactivity of an essential oil derived from Monarda didyma L. (scarlet beebalm). M.didyma is a flowering plant native to North America which has a rich history of use in herbal medicine among Native Americans. Previous phytochemical studies of M. didyma and related species have identified several bioactive small molecules in their essential oils which offer antimicrobial or anti-inflammatory properties. This study expands on previous works by offering an original phytochemical analysis of essential oil extracted from M.didyma aerial parts which were grown and harvested in Central Italy in 2017. They define a chemotype ("Urbino chemotype") for this essential oil, which is predominantly composed of 20 chemical compounds. Many of these compounds are present in other M. didyma chemotypes, but at least two (carvacrol and carvacrol methyl ether) have not been seen previously. In support of the plant’s therapeutic potential, the authors perform in vitro analyses that demonstrate that M.didyma essential oil has free radical-scavenging activity, and that it modulates the transcription of key regulators in the human TLR4 signaling pathway. Based on these findings, the authors conclude that the M.didyma essential oil has intrinsic antioxidant and anti-inflammatory properties which can be explained by its chemical composition. They assert that these findings lend credence to the plant’s utility as a novel resource for naturally derived medicines, food additives, and preservatives.

In general, this manuscript falls within the research scope of PeerJ and conforms to its basic framework. The study’s satisfying mix of chemical, biochemical, and molecular biology analyses provide the manuscript with a multidisciplinary appeal which I believe will be appreciated by the PeerJ audience. However, I think this manuscript could majorly benefit from several editorial revisions which will improve its accessibility and readability. My comments primarily relate to the appropriate presentation of the data, the interpretation of the data, and the broader context of the study’s findings.

Thank you for providing me the opportunity to read your manuscript and learn about your work!

Kristine L. Trotta
University of California, San Francisco
Dept. of Biochemistry and Biophysics

---

## Round 0.2 · Minor Revisions

Dear authors. The second version came with many improvements, however, in the second evaluation, the reviewers, pointed out a few more points for improvement.

So, please, look at all recommendations, and we are waiting for a new version.

Reviewer 1 ·

Basic reporting

1. From the newly added figure 2, how can you differentiate thymol methyl ether and carvacrol methyl ether?

2. 'Q: There is no standard deviation in Table 3, how many repetitions were tested? Except for GC-FID, the repetition time for different chemical analyses were not mentioned. Data presentation in this manuscript is not professional.
A: Standard deviation has been added in New Table 3. All analyses were repeated at least three times.'

Q2: Please include how many times was repeated in each analyses in Materials and Methods. Besides, in Table 3, please include % (weight by weight) or other units.

Experimental design

no comment

Validity of the findings

no comment

·

Basic reporting

-I appreciate the many updates the authors made to the presentation of the data. The addition of the annotated TLC data and NMR spectra are especially helpful.

- In the PDF version of the manuscript, there are some formatting errors that likely arose from the editing process (e.g lines 86, 243; Table 1 legend; Table 3 legend). I wanted to make the authors aware of these as they make their final edits.

- The following figure panels are missing legends: Figure 3b, Figure 5d, Figure 1S, Figure 2S.

- In my original comments, I remarked that supplemental figures 1S and 2S were of low resolution and hard to interpret. I appreciate the authors resubmitting new versions, but they appear to still be low-resolution image files. Perhaps this is a problem on my computer, but I am not sure. In comparison, I found that Figure 3S is of very high resolution and easy to read. It would be helpful for PeerJ to confirm the figure readability before publication.

Experimental design

-The authors sufficiently addressed my comments in this area. Thank you!

Validity of the findings

- I appreciate the effort the authors made to expand their introduction and discussion. Providing the context of related studies in the field helped emphasize the unique goals/outcomes of this study. I particularly appreciated the new sections making comparisons to other M. didyma chemotypes (lines 80; 400).

- I found lines 416-451 to be confusing to read. The density of information is very high, and as a reader I am not sure what compounds to focus on. I think it could be helpful to focus this discussion to just a few abundant compounds that are most biologically interesting across the Italian EO chemotypes.

Additional comments

This manuscript has been significantly improved by the revisions of the authors. In particular, the expanded introduction and discussion help to clarify the motivation for this study and provide helpful context/comparison to other studies in this field. I suggest very minor edits for this version of the manuscript, mostly pertaining to formatting and clarification of some descriptive text. I’d also like to thank the authors for their kind words about my review.

---

## Round 0.3 · accepted · Accept

Congrats. Your manuscript was accepted. Please, note that PeerJ will help you until the publication.

Best regards,

Reviewer 3 ·

Basic reporting

Overall, this manuscript has been significantly improved. The revised introduction clearly addresses the value of this report and how this study differentiates from other previous publications. I appreciated several updates in many figures, the actual image of Monarda didyma L. in figure1, the addition of figure2, figure4 and figure5d as well as the chemical structure of each compound presented in figure3. All these changes enhance the quality of this manuscript and will be helpful for general readers.

Experimental design

Appropriate methods were performed to collectively identify the chemical composition including LC, GC-FID, GC-MS and 1H-NMR. The in vitro anti-inflammatory activity of the essential oil was examined by a proper and relevant model study of LPS stimulated cells. All methods are described with sufficient details and information to replicate.

Validity of the findings

The authors sufficiently addressed my comments in this area. Thank you.

·

Basic reporting

no comment

Experimental design

no comment

Validity of the findings

no comment

Additional comments

no comment